# A Systematic Review of Diagnostic Modalities and Strategies for the Assessment of Complications in Adult Patients with Neurofibromatosis Type 1

**DOI:** 10.3390/cancers16061119

**Published:** 2024-03-11

**Authors:** Sounak Rana, Chen Ee Low, Manasadevi Karthikeyan, Mark Jean Aan Koh, Joanne Ngeow, Jianbang Chiang

**Affiliations:** 1Department of Medicine, Yong Loo Lin School of Medicine, National University of Singapore, Singapore 117597, Singapore; e0886383@u.nus.edu (S.R.); cheneelow@u.nus.edu (C.E.L.); 2Cancer Genetics Service, National Cancer Centre, Singapore 168583, Singapore; manasadevi.karthikeyan@nccs.com.sg; 3KK Women’s and Children’s Hospital, Singapore 229899, Singapore; mark.koh.j.a@singhealth.com.sg; 4Duke-NUS Medical School, Singapore 169857, Singapore; joanne.ngeow@ntu.edu.sg; 5Lee Kong Chian School of Medicine, Nanyang Technological University, Singapore 637551, Singapore; 6Institute of Molecular and Cellular Biology, Agency for Science, Technology and Research, Singapore 138672, Singapore; 7Division of Medical Oncology, National Cancer Centre, Singapore 168583, Singapore

**Keywords:** NF1, genetic testing, genetic counselling, multidisciplinary clinics, tumour predisposition syndrome

## Abstract

**Simple Summary:**

Neurofibromatosis Type 1 is an inherited tumour predisposition syndrome with a varied clinical phenotype. Long-term monitoring through imaging is inconsistent and varies in high- and low-income countries. Implementation of a clinical practice guideline through a multidisciplinary clinic is instrumental to the care of adult Neurofibromatosis Type 1 patients. This systematic review aims to evaluate the association between a country’s socioeconomic status and diagnostic modalities and strategies used for adult Neurofibromatosis Type 1 patients. Our results show multiple imaging modalities are used in high-income countries; however, there is limited use in low-income countries. The two most common diagnostic modalities used in developed countries are WB MRI and FDG PET/CT.

**Abstract:**

Background: Neurofibromatosis Type 1 is an autosomal dominant tumour-predisposition condition commonly diagnosed in childhood and fully penetrant by adulthood. Long-term monitoring through imaging is inconsistent and varies between high- and low-income countries. Implementation of a clinical practice guideline through a multidisciplinary clinic is instrumental to the care of adult Neurofibromatosis Type 1 patients. We aim to systematically review international diagnostic modalities and strategies to evaluate any association between a country’s socioeconomic status and diagnostic modalities or strategies used for Neurofibromatosis Type 1 patients. Methods: We searched PubMed, Embase, Web of Science, and Cochrane. Relevant clinical information on the surveillance of adult Neurofibromatosis Type 1 patients worldwide was reviewed, extracted, and synthesised. Results: We identified 51 papers reporting on 7724 individuals. Multiple imaging modalities are actively employed in high-income and upper-middle-income countries for surveying adult Neurofibromatosis Type 1 patients. We did not find any relevant papers from low- and middle-income countries. Conclusions: This systematic review suggests that there is robust data on diagnostic modalities for adult Neurofibromatosis Type 1 patients in high-income countries, but not for low- and middle-income countries. There is a lack of data on consolidated diagnostic strategies from both high- and low-income countries. Efforts should be made to publish data on usual clinical practice in low- and middle-income countries to develop clinical practice guidelines describing best medical practice to fit a local context.

## 1. Introduction

Neurofibromatosis Type 1 (NF1) is an autosomal dominant hereditary condition caused by germline pathogenic variants in the *NF1* gene. Individuals with NF1 have an 8–15-year reduction in lifespan due to malignant neoplasms, such as breast cancer, malignant peripheral nerve sheath tumours (MPNSTs), pheochromocytomas, and cardiovascular complications caused by essential hypertension and renal artery stenosis [1].

Per the revised diagnostic criteria for NF1, the condition is characterised by café-au-lait macules (CALMs), freckling in the axillary or inguinal region, neurofibromas and plexiform neurofibromas, optic pathway gliomas (OPGs), iris Lisch nodules, osseous lesions including sphenoid dysplasia, anterolateral tibial bowing, pseudoarthrosis of a long bone, and a heterozygous pathogenic *NF1* variant [2]. Clinical diagnosis of the condition is made shortly after birth and during early childhood, usually by age 10 [3]. As patients mature to adolescence and adulthood, several complications arise, such as an increased risk of malignancy, cutaneous, ocular, orthopaedic, and vascular symptoms [4].

### Aims of the Paper

NF1 management requires a multidisciplinary approach as outlined in consolidated diagnostic strategies [4,5,6,7]. Local requirements may result in different diagnostic strategies across high- and low-income countries. Diagnostic modalities for complications in NF1 patients include regular history and physical examinations, with consideration for baseline scanning—typically whole-body magnetic resonance imaging (WBMRI) and the assessment of clinically suspicious lesions via fluorodeoxyglucose positron emission tomography/computerised tomography (FDG PET/CT), and other relevant imaging modalities. This testing allows for the evaluation of internal tumour burden [8], setting a baseline to compare future growth and enable the assessment of malignant change [9]. The assessment of suspicious lesions, such as malignant peripheral nerve sheath tumours (MPSNTs) causing new-onset limb weakness or progressive unremitting pain [10], or other soft tissue sarcomas, often require imaging modalities including MRI, FDG PET/CT, and computerised tomography (CT) scans. The usage of CT, FDG PET/CT, and MRI are limited in low- and lower-middle-income countries due to high acquisition costs, the need for relevant infrastructure, and trained staff for operation, maintenance, and interpretation [11]. As a result, there may be significant differences in both diagnostic strategies and modalities used for the management of adult NF1 patients between high- and lower-middle-income countries.

This paper aimed to systematically review international diagnostic modalities and strategies to evaluate any association between a country’s socioeconomic status and differences in diagnostic modalities and strategies used for NF1 patients.

## 2. Methods

### 2.1. Protocol and Guidance

The systematic review is reported according to the Preferred Reporting Items for Systematic Reviews and Meta-Analyses (PRISMA) guidelines [12]. Our protocol is registered on PROSPERO (Reference: CRD42023428068).

### 2.2. Data Sources and Search Strategy

A literature search was performed in PubMed, Embase, Web of Science, and Cochrane. The search strategy combined search terms for NF1, surveillance, and adults. Subject headings were searched using database-controlled vocabulary. Various synonyms, truncated appropriately, were used to search for title, abstract, and author keywords. Due to rapidly evolving NF1 diagnostic strategies, the search was limited to publications from 2000 to the current day. The search strategy was translated between each database. The full strategies for PubMed and EMBASE are available in Appendix A.

### 2.3. Definitions

A diagnostic strategy is defined as a systematic plan for the monitoring of adult NF1 patients to detect and address possible medical complications and disease progression. Diagnostic modalities are defined as investigations used to assess and survey clinical manifestations in adult NF1 patients (Appendix A).

### 2.4. Study Selection: Inclusion and Exclusion Criteria

Two reviewers independently screened the titles and abstracts of all studies to determine eligibility in accordance with the inclusion and exclusion criteria. Studies in which eligibility was unclear were independently evaluated by two other reviewers. Disparities were settled via adjudication by a third senior reviewer.

We included peer-reviewed studies in English published since 2000 that examined diagnostic modalities for adult NF1 patients. Adult NF1 patients are defined as participants who are at least 18 years of age with a clinical or genetic diagnosis of NF1. We included studies that reported on the use of surveillance methods to diagnose clinical pathologies and monitor disease progression in NF1 patients. Case reports, grey literature, and non-English articles were excluded due to difficulty of interpretation owing to a language barrier, and studies that did not stratify NF1 patients by age were excluded. Of the two non-English articles excluded, one was in Mandarin and the other in French.

We used published research as a proxy to estimate clinical practice in other countries, based on the assumption that there is a direct correlation between publications utilising diagnostic modalities and the clinical use of diagnostic modalities.

### 2.5. Data Extraction and Organisation

The subject matter information included country of study, demographics of NF1 patients, and the surveillance modality used. For data related to country of study, all data were extracted and classified according to whether the country was a (A) low-income economy (gross national income (GNI) per capita of USD 1085 or less), (B) lower-middle-income economy (GNI per capita from USD 1086 to USD 4255) (C) Upper-middle-income economy (GNI per capita from USD 4256 to USD 13,205) (D) High-income economy (GNI per capita of USD 13,205 or more) according to the definitions set by the World Bank [13] and using GNI per capita data from the World Bank [14].

### 2.6. Risk of Bias Assessment

We assessed the methodological quality and risk of bias of the included studies via the Joanna Briggs Institute (JBI) critical appraisal checklist [15], which comprises appraisal of the inclusion criteria, measurement of condition, reporting of baseline characteristics, reporting of outcomes, and appropriateness of the statistical analysis (Appendix A). This appraisal was performed by two independent reviewers with disparities adjudicated by a third senior reviewer.

## 3. Results

A total of 51 studies was included with the screening process outlined in Figure 1. An overview of the studies is provided in Table 1.

### 3.1. Diagnostic Modalities

Six types of diagnostic modalities—MRI, FDG PET/CT, CT, physical examination, blood pressure measurement, and bone densitometry—were used in the included studies (Table 2). Table 2 demonstrates the total number of studies, categorised by NF1 complication assessed, and the total number of imaging studies described by our included papers, categorised by the type of imaging modality. Of the included studies, 34 (50.7%) used MRI, 16 (23.8%) employed FDG PET/CT, 5 (7.5%) utilised CT, 7 (10.4%) implemented physical examination, 2 (3.0%) involved blood pressure measurement, and 3 (4.5%) used bone densitometry scans.

Seven studies involved physical examination; three studies involved a dermatological examination for CALMs, skinfold freckling, cutaneous and subcutaneous neurofibromas [21,62,64], four involved a full ophthalmologic exam for symptomatic OPGs and slit lamp examination for Lisch nodules [20,21,63,64], and three involved a neurologic exam [20,21,26].

Two studies involved blood pressure measurements to aid screening for headaches [26,64] and as part of a routine physical examination [64].

Thirty-four studies involved the use of MRI, of which seven assessed internal tumour burden [18,34,36,37,38,41,42] and eleven involved the assessment of MPNSTs [20,24,25,27,31,35,40,57,58,59,66]. Five studies involved the assessment of spinal abnormalities or lesions [16,23,29,32,39]. Two studies involved the assessment OPGs [28,44], while two involved the assessment of extra-optic pathway gliomas [28,43]. Neurovascular complications [30], headache [26], UBOs [17], epilepsy [33], cognitive function [19], plexiform neurofibromas [21,62], and bone health [62] were assessed by one study each.

Sixteen studies involved the use of FDG PET/CT, of which all involved assessment of MPNSTs [45,46,47,48,49,50,52,53,54,55,56,57,58,59,66,67].

Furthermore, five studies involved the use of CT scans; three involved a multi-detector CT (MDCT) for the evaluation of spinal abnormalities [16], GISTs [65] and diffuse interstitial lung disease [45] respectively. Two studies involved CT scans to aid diagnostic imaging of tumours, including MPNSTs [35], non-optic pathway gliomas, and OPGs [44].

Lastly, three studies involved bone densitometry for the assessment of bone mineral density to diagnose osteopenia or osteoporosis [60,61,62].

### 3.2. Socioeconomic Status

Fifty studies were conducted in high-income countries and one was conducted in an upper-middle income country—Türkiye (Table 2). None were conducted in low-income or lower-middle-income countries. The high-income country group included Germany (14/51, 27.5%) [9,19,20,21,22,28,31,39,45,47,50,51,52,60], France (8/51, 15.7%) [35,40,44,46,53,56,63,64], United Kingdom (7/51, 13.7%) [26,29,30,32,43,49,58], other European countries (3/51 5.9%) [38,61,62], the United States (14/51, 27.5%) [16,18,23,27,33,34,36,37,41,42,48,54,57,59], and Japan (4/51, 7.8%) [24,25,55,65].

Multiple studies employed more than one surveillance modality. Of the papers from Germany, eight employed MRI, six used FDG PET/CT, one utilised CT, two utilised physical examinations, and one implemented a bone densitometry scan. In comparison, of the papers from the United States, 12 used MRI, 4 employed FDG PET/CT, and 1 utilised a CT scan. In contrast, of the papers from France, three employed MRI, three used FDG PET/CT, two implemented CT, three involved physical examination, and just one applied blood pressure measurement. Within the papers from the United Kingdom, six involved the use of MRI, two implemented FDG PET/CT, one utilised physical examination, and just one employed blood pressure measurement. Additionally, of the papers from Japan, two used MRI, one utilised FDG PET/CT, and one employed a CT scan. One study from Italy used MRI, physical examination, and bone densitometry scans. The single study from Finland employed bone densitometry scans, while the single studies from Belgium and Türkiye each utilised MRI scans.

### 3.3. Outcomes

In our included studies, FDG PET/CT was primarily utilised for the assessment of MPNSTs. Table 3 shows the number of patients, number of lesions, quantified parameter, threshold value, maximum standardised uptake value (SUVmax), sensitivity, and specificity in the 12 studies included in the analysis. Of the studies included in the analysis, the mean SUVmax values of MPNSTs and benign neurofibromas were 7.98 and 2.93, respectively. A wide range of optimum SUVmax values to maximise sensitivity and specificity is noted, with threshold values ranging from 3.0 to 4.5. Sensitivities ranged from 75% to 100%, and specificities ranged from 68.9% to 100%.

Bone densitometry was used to assess the prevalence of osteopenia and osteoporosis in NF1 patient populations. Table 4 shows the total number of patients, the number of patients with normal bone mineral density (BMD), the number of patients with osteopenia, and the number of patients with osteoporosis in the three included studies. Of the studies included in the analysis, the mean prevalence of osteopenia and osteoporosis was 35.6% (range = 26.3–42.9%), and 37.2% (range = 12.5–57.1%) respectively.

### 3.4. Risk of Bias

The quality of the methodology in the 51 studies included in systematic review, as scored via the JBI checklist, is presented in Appendix A. Overall, the results indicated a low risk of bias for most studies and a moderate risk of bias for six studies [20,21,48,54,64,66], primarily due to a lack of clearly stated confounding factors, strategies to deal with confounding factors, and due to the participants not being free of the outcome at the start of the study.

## 4. Discussion

Of the 51 included studies, none reported on consolidated diagnostic strategies. Fifty-one papers reported on the efficacy of various diagnostic modalities, namely MRI FDG PET/CT (Table 3) and bone densitometry (Table 4), in the assessment of tumour burden, the diagnosis of MPNSTs, and the evaluation of osteoporosis in adult NF1 patients, respectively. Their results substantiate and inform screening recommendations outlined in several guidelines from high-income countries, namely the ACMG, ERN GENTURIS, and the national French and British guidelines. In turn, parts of the aforementioned guidelines may be adapted for use in the local setting of low-income and lower-middle-income countries. Efforts should be made to encourage the publication of consolidated diagnostic strategies from various countries, informing the development of effective, cost-sensitive strategies for use in low-income and lower-middle-income countries.

From the results of our systematic review, various diagnostic modalities are employed in the surveillance of adult NF1 patients, including imaging modalities—MRI, FDG PET/CT, CT, and bone densitometry—measurement of vitals—blood pressure monitoring—and physical examinations with history-taking.

The two most frequently used diagnostic modalities were MRI and FDG PET/CT scans, which are the primary choices for baseline imaging for NF1. Internal neurofibroma burden is a common clinical manifestation of NF1 [8], and high internal neurofibroma burden is often associated with an elevated risk of MPNSTs [40]. The establishment of a baseline enables serial monitoring and assessment of internal neurofibroma burden and flags up sudden growth associated with malignant transformation.

The US had the highest number of studies involving MRI. This was significantly greater than those of the UK and European Union countries such as France and Germany. Several factors may underlie this finding. Firstly, this may reflect differences in best clinical practice outlined by guidelines published in the respective countries (Table 5). The American College of Major Genetics and Genomics (ACMG) has published a clinical practice resource for medical geneticists and other clinicians for the care of adults with NF1. ACMG recommends a baseline MRI of known or suspected non-superficial plexiform neurofibromas as a gauge of tumour burden, as part of the clinical assessment of adults with NF1 [4]. Members of the United Kingdom Neurofibromatosis Association Clinical Advisory Board provided a clinical practice guideline for diagnosis and management of NF1 [1]. In contrast, this set of guidelines from the UK does not recommend baseline brain and spine MRI for the monitoring of asymptomatic tumours. Similarly, the ERN GENTURIS tumour surveillance guidelines developed for Europe, primarily addressing the clinical assessment and imaging screening of NF1-related tumours in adolescents and adults [5], provides only a weak recommendation for WBMRI in the monitoring of plexiform neurofibromas. The national French guidelines on NF1 management were developed in accordance with the French national plan for rare diseases, intended to provide healthcare professionals with guidelines regarding the diagnosis and therapeutic management of NF1 [6]. The French guidelines do not systematically recommend WBMRI for the detection of internal neurofibromas or MPNSTs in the screening of adult NF1 patients. Therefore, practice of assessing tumour burden via MRI scans outlined in the studies from the US may reflect its recommendation in the country’s clinical practice guidelines. Secondly, the frequency of MRI use in studies from the US may reflect the high availability of MRI units in the country. As of 2021, the US has 38.0 MRI machines per million people [68], compared to 35.3 in Germany, 17.0 in France, and 7.4 in the UK [68]. The relative abundance of MRI units in the US may increase the feasibility of using routine MRI scans for the monitoring of adult NF1 patients. This partially explains its inclusion in guidelines, and the higher frequency of its use in studies from the US. Thirdly, a possible reason for the difference in MRI use may be because a larger proportion of the US population is covered under private health insurance, as opposed to the systems of universal healthcare instituted in Germany, France, and the UK, which may enable more intensive use of expensive MRI scans. Angell et al. suggested that the private healthcare insurance system in the US may incentivise MRI usage [69]. According to the Congressional Research Service, as of 2021, 68.4% of Americans were privately insured, as opposed to 39.3% who were covered under federal insurance programs, namely, Medicare and Medicaid [70]. In addition, many healthcare providers also provide a fee-for-service model, where private insurance providers are billed in accordance with the procedure ordered, incentivising use of expensive imaging modalities such as MRI [71]. In contrast, the Social Security insurance program in France covers all residents [72], like the universal healthcare provided by the National Health Service (NHS) in the UK [73] and the statutory health insurance in Germany, which covers 88% of all residents. MRI units may be considered a common resource in countries with universal healthcare coverage, and the need for more efficient allocation of resources to patients and stricter control of expenditures may discourage its recommendation in guidelines for adult NF1 patients, and its use within our collected studies from Europe as compared to the US.

European countries—namely Germany and France—had a higher proportion of studies than the US or the UK involving FDG PET/CT for assessing MPNSTs. There are several key differences between clinical practice guidelines from the respective countries (Table 5). When clinically suspicious, the ERN GENTURIS guidelines recommend FDG PET/CT for the identification of MPNSTs in adults with NF1. Similarly, the French guidelines recommend FDG PET/CT as the most sensitive and specific indicator of malignant potential for MPNSTs. In contrast, the ACMG guidelines recommend targeted MRI scanning for the early detection of MPNSTs in settings of clinical suspicion. The difference may be due to differing guidelines in use of FDG PET/CT or MRI for surveillance in adult NF1 patients across the Atlantic. This explains the lower proportion of studies from the US including FDG PET/CT in our dataset. Although the UK guidelines promote FDG PET/CT for the visualisation of MPNSTs, other systemic factors may hamper its use. The UK has fewer PET/CT machines, 0.08 per 100,000 people [74], in comparison to 0.12 in Germany [75]. The relative scarcity of PET/CT scanners in the UK may underlie the comparative lower proportion of studies involving PET/CT scans from the UK, as shown in our study.

None of the 51 studies reporting on diagnostic modalities were from low-income and lower-middle-income countries. In addition, there is a paucity of published clinical practice guidelines from most low-income and lower-middle-income countries, with the exception of a guideline published in Brazil [76]. Therefore, there is an overall inadequacy of information regarding ongoing clinical practice, and guidelines regarding best clinical practice in the management of adult NF1 in these countries. In turn, it is not tenable for low-income and lower-middle income countries to adhere to guidelines set by high-income countries such as those by ERN GENTURIS or ACMG, as the ongoing surveillance of adult NF1 requires intense usage of expensive modalities, namely MRI and FDG PET/CT. The Brazilian guidelines suggest an annual screening regimen comprising an ophthalmologic exam, a physical exam of the skin and skeleton, an audiological and speech evaluation, cognitive testing, oral evaluation, arterial blood pressure measurement, and an electroencephalogram—all low-cost and accessible investigations. However, the guidelines also call for the use of FDG PET/CT or MRI when there is clinical suspicion of the malignant transformation of a plexiform neurofibroma. It should be recognised that Brazil had just 118 PET/CT scanners and 3174 MRI machines for a population of more than 203 million as of 2022, making the postulated best medical practice highly infeasible for many adult NF1 patients [77]. There is a globally inequitable distribution of medical imaging, and high-cost modalities for NF1 surveillance such as MRI and FDG PET/CT have many barriers to access in low- and middle-income economies [78], making their widespread employment in adult NF1 surveillance implausible. Such barriers may include a lack of investment plans or prioritisation, high costs associated with equipment procurement, maintenance, and safety, and the requirement of properly trained technicians for operation of imaging equipment [79]. Education and upskilling of the workforce to facilitate the operation of imaging equipment [79] is vital to improve access. Besides global inequities in imaging access, regional inequities in access may also lead to the neglect of underserved populations. Karia et al. found that patients from lower socioeconomic areas within the UK had a comparative lack of access to diagnostic neuroimaging facilities [80]. Therefore, collaboration between high- and low- to middle-income countries to develop guidelines to fit their means and local context more accurately, and to publish more data locally regarding the surveillance and outcomes of adult NF1 patients, is warranted.

A limitation of our study is the absence of an analysis or comparison between the methodologies of studies involving the same imaging modalities. Further studies examining the differences in image acquisition and processing between papers involving surveillance imaging may provide valuable insight. Another limitation, as previously mentioned, is the absence of studies reporting on diagnostic strategies for adult NF1 patients despite our stated aim to investigate both diagnostic modalities and strategies of various countries. One other limitation is the exclusion of non-English articles due to technical limitations in the interpretation of the data, owing to a language barrier. The language of the article may be a confounder, as articles written in English may involve surveillance techniques more often than those in other languages.

As a future direction, regional foundations familiar with the local context of low-income and lower-middle-income countries can collaborate with well-established international organisations such as the Children’s Tumour Foundation. This may allow the development of resource-sensitive diagnostic strategies stratified by GNI and suited to the cultural and socioeconomic context of low-income and lower-middle-income countries.

## 5. Conclusions

This systematic review suggests that there is robust data on diagnostic modalities for adult NF1 patients in high-income countries, but not for lower-income countries. Low- to middle-income countries may lack the financial resources, technical expertise, and access to advanced imaging modalities to adhere to guidelines set by high-income countries, and therefore a collaborative effort should be made to publish usual clinical practice pertaining to the surveillance of adult NF1 patients, to help develop a clinical practice guideline that best fits their local needs and context to ensure equitable management of NF1 patients. In addition, in our included studies, there is a lack of data on consolidated diagnostic strategies from both low- and high-income countries. Efforts should be made to publish diagnostic strategies in both low- and high-income countries. Diagnostic strategies promulgated in low-income countries may provide a reference for practical, grounded approaches and foster peer-to-peer collaboration in the development of cost-effective adult NF1 diagnostic strategies, whereas those published in high-income countries may set a standard to strive towards. Additionally, based on our systematic review, we recommend the inclusion of MRI and FDG PET/CT scans for the assessment of tumour burden and detection of MPNSTs, respectively, for adult NF1 surveillance. This is based on its wide inclusion in accepted clinical practice guidelines and its frequent utilisation in multiple high-income countries as standard practice.

## Figures and Tables

**Figure 1 cancers-16-01119-f001:**
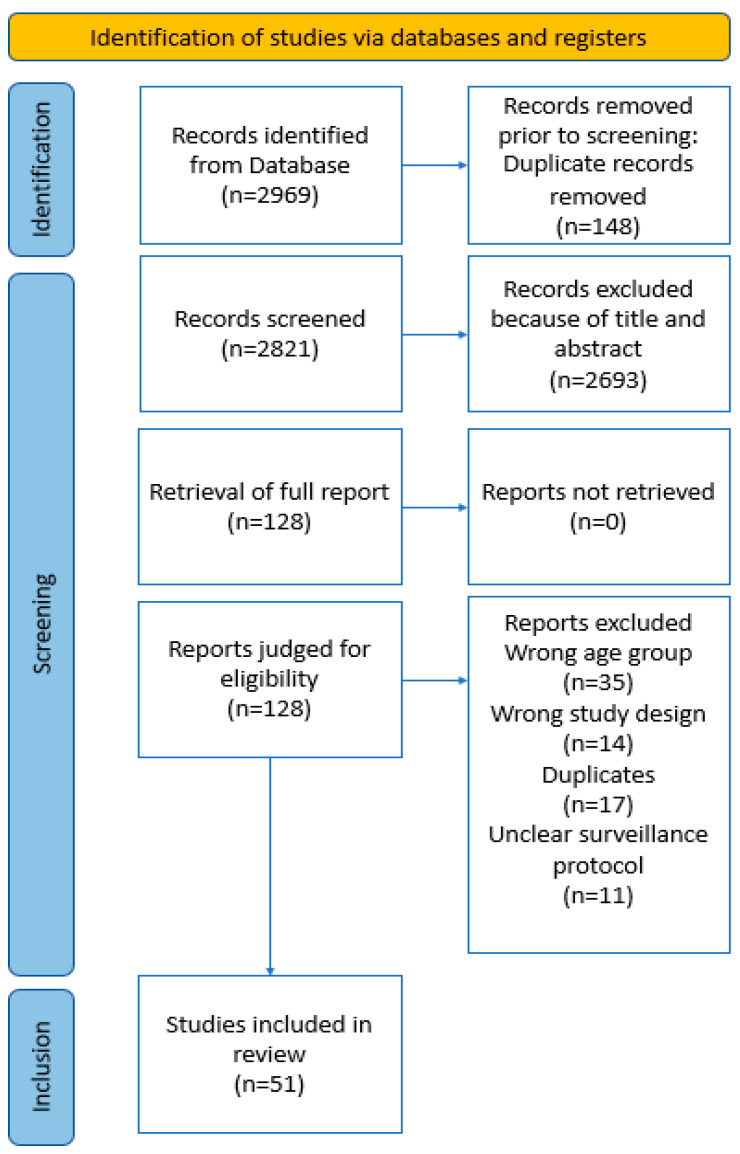
PRISMA flowchart.

**Table 1 cancers-16-01119-t001:** Details of 51 studies involving diagnostic modalities grouped by imaging modality.

	Author	Year	Country (GNI per Capita)	Other Modalities Included	Age (Mean (SD))	Number of Subjects	Pathology	Study Methodology and Characteristics
Studies Involving MRI as a Modality								
1	Debnam [16]	2014	United States (>USD 13,205)	MDCT, plain radiographs	41 (16)	73	Spinal abnormalities	A cohort study involving NF1 patients who underwent MDCT scans for diagnostic, preoperative, or postoperative evaluation of spinal abnormalities. The study aimed to investigate how MDCT could be used for this patient population.
2	Alkan [17]	2005	Turkey (USD 4256 to USD 13,205)	-	23 (15)	30	UBOs	A case-control study of NF1 patients who underwent MRI scans. The study aimed to evaluate the differences in ADC values between infra- and supratentorial UBOs in NF1 patient and control groups and to investigate the correlation between age and ADC values.
3	Cai [18]	2009	United States (>USD 13,205)	-	42 (15)	28	Tumour burden	A cohort study of NF1 patients who underwent MRI scans. The study aimed to develop a 3D segmentation and computerised volumetry modality for use in the assessment of neurofibromatosis and to assess the ability of this modality to aid in the calculation of tumour burden in NF1 patients.
4	Feldmann [19]	2003	Germany (>USD 13,205)	-	16 (9)	200	Motor and cognitive function	A case-control study of NF1 patients and age/sex/socioeconomic status-matched controls who underwent MRI scans of the brain. The study aimed to examine intelligence and school career, motor performance, and cranial MRI in a large series of 100 children and young adults with NF1.
5	Friedrich [20]	2005	Germany (>USD 13,205)	Ultrasound, physical examination (ophthalmological and neurological)	34 (12)	10	MPNST	A cohort study of NF1 patients who underwent brain MRI, ultrasound, and physical examinations. The study aimed to compare the mutation type and MR imaging characteristics of MPNSTs.
6	Mautner [21]	2006	Germany (>USD 13,205)	Physical examination (dermatological, ophthalmological and neurological)	28 (−)	202	PNFs	A cohort study of NF1 patients who underwent MRI and physical examinations. The study aimed to characterise growth patterns of PNFs and associated disfigurement and functional deficits.
7	Sellmer [22]	2017	Germany (>USD 13,205)	-	NR	562	Non-optic gliomas	A cohort study of NF1 patients who underwent brain MRIs. The study aimed to investigate progression, spontaneous regression, and the natural history of non-optic gliomas in adults and compare these findings to the results found in children.
8	Jaremko [23]	2012	United States (>USD 13,205)	-	41 (−)	141	Scoliosis and other incidental findings	A cohort study of NF1 patients who underwent WBMRI. The study aimed to demonstrate incidental findings and scoliosis on WBMRI in patients with NF1.
9	Koike [24]	2022	Japan (>USD 13,205)	-	35 (16)	30	MPNST	A cohort study of NF1 patients with PNFs or MPNSTs who underwent MRI. The study aimed to investigate the usefulness of DWI in differentiating PNFs and MPNST in NF1 patients.
10	Matsumine [25]	2009	Japan (>USD 13,205)	-	43 (−)	37	MPNST	A cohort study of NF1 patients with neurogenic tumours in the extremities or on the trunk who underwent MRI. The study aimed to define the criteria for the differential diagnosis between NF and MPNST on MRI in NF1.
11	Afridi [26]	2015	United Kingdom (>USD 13,205)	Blood pressure measurement, physical examination (neurological)	39 (10)	115	Headache	A cohort study of NF1 patients who received blood pressure measurements, neurological examinations, and MRI scans. The study aimed to characterise the phenotype and prevalence of headaches in NF1 patients and determine the quality-of-life impact.
12	Chhabra [27]	2011	United States (>USD 13,205)	-	37 (18)	56	MPNST	A cohort study of NF1 patients with diagnoses of MPNST/BPNSTs who underwent MRI scans. The study aimed to determine the potential differentiating MRI features between NF-MPNSTs and non-NF-MPNSTs and between BPNSTs and MPNSTs.
13	Sellmer [28]	2018	Germany (>USD 13,205)	-	NR	562	Optic pathway gliomas	A cohort study of NF1 patients who were offered WB and head MRIs. The study aimed to determine the natural history of OPGs in children and adults with NF1.
14	Curtis-Lopez [29]	2020	United Kingdom (>USD 13,205)	Plain radiograph	38 (−)	303	Spinal lesions	A cohort study of NF1 patients with radiological evidence of spinal lesions seen on MRI spine or spinal X-ray. The study aimed to report the prevalence of different spinal lesions and attempt comparisons between the spinal phenotype and classic NF1 groups.
15	Sheerin [30]	2022	United Kingdom (>USD 13,205)	-	NR	2068	Neurovascular complications	A cohort study of NF1 patients who received cranial MRI imaging or were diagnosed with various neurovascular complications. The study aimed to assess the frequency and clinical and imaging spectrum of vascular complications in an adult cohort of NF1 patients.
16	Salamon [31]	2019	Germany (>USD 13,205)	-	34 (−)	26	MPNST	A cohort study of NF1 patients with clinical suspicion of MPNST who received DW-MRI. The study aimed to determine the value of DW-MRI for the characterisation of BPNSTs and MPNSTs in NF1 patients.
17	Ramachandran [32]	2004	United Kingdom (>USD 13,205)	Plain radiograph	16 (−)	27	Spinal deformity	A cohort study of NF1 patients who underwent WSMRI and plain radiographs. The study aimed to determine the role of whole-spine MRI in the classification and management of patients with NF-1 and spinal deformity.
18	Pecoraro [33]	2017	United States (>USD 13,205)	-	18 (16)	184	Epilepsy	A cohort study of NF1 patients with epilepsy/epilepsy syndromes who underwent MRI. The study aimed to describe the characteristics of epilepsy in patients with NF1.
19	Heffler [34]	2017	United States (>USD 13,205)	-	42 (14)	15	Tumour burden	A cohort study of NF1 patients who underwent WBMRI with segmentation. The study aimed to perform segmentation of WBMRI to assess the feasibility and quantitate the total tumour volume (tumour burden) in NF1 patients and examine associations with demographic, disease-related and anthropomorphic features.
20	Tucker [35]	2005	France (>USD 13,205)	Physical examination (ophthalmologic), plain radiograph, ultrasound, CT, 24 h urinary catecholamine analysis	35 (13)	476	MPNST	A cohort study of NF1 patients who underwent routine clinical assessment. The study aimed to determine whether NF1 patients who have benign neurofibromas of various kinds are at greater risk of developing MPNSTs than patients with NF1 who lack these benign tumours.
21	Ly [36]	2023	United States (>USD 13,205)	-	43 (12)	47	Internal neurofibromas	A cohort study of NF1 patients with a baseline WBMRI who underwent a follow-up WBMRI. The study aimed to evaluate the long-term growth behaviour of internal neurofibromas in adults with NF1.
22	Well [37]	2020	United States (>USD 13,205)	-	25 (6)	13	Neurofibroma growth during pregnancy	A case-control study of pregnant NF1 patients and an age-matched female control group. The study aimed to quantify the growth of cutaneous and plexiform neurofibromas in NF1 patients during pregnancy and to assess NF1-related clinical symptoms.
23	Van Meerbeeck [38]	2009	Belgium (>USD 13,205)	-	37 (11)	24	Neurofibromas	A cohort study of NF1 patients who underwent WBMRI. The study aimed to assess the value of WBMRI in NF1 patients.
24	Well [39]	2021	Germany (>USD 13,205)	-	27 (−)	537	Spinal abnormalities	A case-control study of NF1 patients and age/sex-matched non-NF1 patients who underwent WBMRI. The study aimed to quantify the prevalence of spinal abnormalities in NF1 patients, associate the co-appearance of spinal abnormalities with both NF1 and clinical symptoms, and investigate whether different mutations of the NF1 gene affect the prevalence of these abnormalities.
25	Mautner [40]	2008	France (>USD 13,205)	-	31 (18)	39	MPNST	A case-control study of NF1 patients with MPNSTs and controls without MPNST who underwent WBMRI. The study aimed to evaluate the relationship of the total body burden of internal neurofibromas to MPNSTs.
26	Plotkin [41]	2012	United States (>USD 13,205)	-	39 (−)	247	Internal tumour burden	A cohort study of neurofibromatosis patients who underwent WBMRI. The study aimed to establish an international cohort of patients with quantified whole-body internal tumour burden and to correlate tumour burden with clinical features of the disease.
27	Zhang [42]	2017	United States (>USD 13,205)	-	39 (14)	30	Internal tumour burden	A cohort study of NF1 patients who underwent WBMRI and WSMRI. The study aimed to determine the incremental value of multiparametric WBMRI over WSMRI in NF1 patients.
28	Byrne [43]	2017	United Kingdom (>USD 13,205)	-	16 (−)	100	Non-optic pathway gliomas	A cohort study of NF1 patients with a history of non-OPG who underwent serial neuroimaging. The study aimed to characterise the clinical presentation, management, progression, and outcomes in a non-OPG NF1 cohort.
29	Guillamo [44]	2003	France (>USD 13,205)	CT	32 (9)	16	Optic pathway and extra optic pathway CNS tumours	A cohort study of NF1 patients with presence of a CNS tumour who underwent imaging. The study aimed to identify prognostic factors for patients with CNS tumours and NF1.
Studies Involving FDG PET/CT as a Modality								
1	Avanesov [45]	2021	Germany (>USD 13,205)	MDCT	33 (14)	71	MPNST	A cohort study involving NF1 patients who had undergone 18 FDG PET/CT for exclusion of MPNST, who received MDCT for assessment of lung manifestations. The study aimed to evaluate the smoking history, patients’ age, genetics, and the presence of MPNSTs as potential influencing factors for lung pathologies.
2	Brahmi [46]	2015	France (>USD 13,205)	Biopsy	35 (12)	26	MPNST	A cohort study involving NF1 patients with a clinical suspicion of MPNST and a suspect lesion from a PET/CT scan, who received PET/CT-guided percutaneous biopsy for a pathological diagnosis. The study aimed to investigate the effectiveness of, and complications associated with PET/CT-guided percutaneous biopsies for an NF1-related MPNST diagnosis.
3	Brenner [47]	2006	Germany (>USD 13,205)	-	32 (12)	16	MPNST	A cohort study involving NF1 patients with MPNSTs who underwent PET imaging with FDG during routine preoperative staging, and underwent wide local excision of the tumour which was then histologically examined. The study aimed to evaluate the potential of 18F-fluorodeoxyglucose positron emission tomography (FDG PET) for the prediction of patient outcomes in MPNST.
4	Chirindel [48]	2015	United States (>USD 13,205)	-	39 (16)	41	MPNST	A cohort study involving NF1 patients presenting new symptoms or enlarging lesions, who were clinically evaluated with early and delayed 18F-FDG PET/CT imaging. The study aimed to compare the effectiveness of early (1 h) and delayed (4 h) 18F-FDG PET/CT imaging in differentiating MPNSTs from BNFs.
5	Cook [49]	2017	United Kingdom (>USD 13,205)	-	35 (−)	54	MPNST	A cohort study involving NF1 patients with clinical suspicion of MPNSTs who received F-18-FDG PET scans. The study aimed to determine whether measurements of 18F-FDG heterogeneity could improve the differentiation of BNFs and MPNSTs.
6	Salamon [50]	2013	Germany (>USD 13,205)	-	33 (15)	50	MPNST	A cohort study involving NF1 patients who had been referred for an 18F-FDG PET/CT scan for exclusion of MPNST and underwent an 18F-FDG PET/CT scan. The study aimed to evaluate the potential usefulness of intratumoural tracer uptake heterogeneity on F-18-FDG PET/CT as compared to SUVmax for PNST characterisation.
7	Salamon [51]	2015	Germany (>USD 13,205)	-	37(12)	36	MPNST	A case-control study involving NF1 patients with MPNSTs and age/sex-matched patients with BPNSTs who underwent F-18-FDG PET/CT scans. The study aimed to determine WB-MTV via F-18-FDG PET/CT and compare WB-MTV between patients with BPNSTs and MPNSTs.
8	Salamon [52]	2014	Germany (>USD 13,205)	-	33 (15)	49	MPNST	A cohort study involving NF1 patients who underwent F-18-FDG PET/CT for exclusion of MPNSTs. The study aimed to evaluate the usefulness of normalising intratumour tracer accumulation on F-18-FDG PET/CT to reference tissue uptake for characterisation of PNSTs compared with the established SUVmax cutoff of >3.5.
9	Van Der Gucht [53]	2016	France (>USD 13,205)	-	33 (12)	49	MPNST	A cohort study involving NF1 patients with clinical suspicion of MPNST who underwent F-18-FDG PET/CT scans. The aim of the study was to investigate the diagnostic and prognostic performances of 18F-FDG PET/CT measures of metabolic tumour burden in NF1 patients with suspicion of malignant transformation.
10	Bredella [54]	2007	United States (>USD 13,205)	-	37 (−)	45	MPNST	A cohort study involving NF1 patients with clinical suspicion of MPNST who underwent F-18-FDG PET scans. The aim of the study was to investigate the use of PET scans in the diagnosis of MPNSTs in NF1 patients.
11	Nishida [55]	2021	Japan (>USD 13,205)	F-18-FDG PET/CT	32 (13)	35	MPNST	A cross-sectional/longitudinal study of NF1 patients with clinical suspicion of tumour presence who received F-18-FDG PET/CT scans. The study aimed to re-confirm the usefulness of PET/CT in the differentiation of benignity/malignancy of neurogenic tumours in NF1 patients and to analyse the natural course of PNFs and clarify whether PET/CT is also useful for detecting non-neurogenic tumours.
12	Combemale [56]	2014	France (>USD 13,205)	F-18-FDG PET	35 (15)	113	MPNST	A cohort study of NF1 patients with suspected MPNSTs who received F-18-FDG PET scans. The study aimed to evaluate a semi-quantitative index for the reproducible detection of MPNST with FDG PET.
13	Ahlawat [57]	2019	United States (>USD 13,205)	MRI	30 (−)	20	MPNST	A cohort study of NF1 patients with PNSTs who received MRI or F-18-FDG PET/CT. The study aimed to determine the utility of quantitative metrics obtained from fMRI using DWI/ADC mapping compared with F-18-FDG PET/CT imaging in NF1 patients for the characterisation of BPNST/MPNST.
14	Derlin [9]	2003	Germany (>USD 13,205)	WBMRI	30 (15)	31	MPNST	A cohort study of NF1 patients referred for the exclusion of MPNST who underwent MRI and F-18-FDG PET/CT. The study aimed to compare the diagnostic performance of F-18-FDG PET/CT and WBMRI for MPNST detection in NF1 patients.
15	Ferner [58]	2000	United Kingdom (>USD 13,205)	MRI	30 (16)	18	MPNST	A cohort study of NF1 patients with clinical suspicion of MPNST who underwent MRI and F-18-FDG PET scans. The study aimed to evaluate the ability of F-18-FDG PET to detect MPNSTs in NF1 patients.
16	Broski [59]	2016	United States (>USD 13,205)	MRI	38 (16)	38	MPNST	A cohort study of NF1 patients with BPNSTs or MPNSTs who underwent F-18-FDG PET and MRI. The study aimed to compare 18F-FDG PET/CT and MRI for differentiating BPNSTs and MPNSTs and correlating imaging characteristics with histopathology.
Studies Involving DXA Scan								
1	Seitz [60]	2010	Germany (>USD 13,205)	Bone densitometry (DXA scan)	47 (16)	56	Bone health	A case-control study involving NF1 patients and age/sex-matched controls who underwent DXA osteodensitometry. The study aimed to perform a systematic clinical and histomorphometric analysis of decreased BMD and vitamin D deficiency in NF1 patients.
2	Heervä [61]	2013	Finland (>USD 13,205)	Bone densitometry (DXA scan)	46 (18)	19	Bone health	A longitudinal study involving NF1 patients, who had initial BMD measurements taken in a 1999 study, and who then underwent DXA scans to measure BMD, 12 years after the initial study. The study aimed to reach the 35 NF1 patients examined in 1999 and assess their bone health after 12 years.
3	Modica [62]	2023	Italy (>USD 13,205)	MRI, physical examination, biochemical testing	41 (11)	31	Bone health	A cross-sectional, case-control study involving NF1 patients and sex/age/BMI- matched controls who underwent bone densitometry (DXA scan), Physical examination, MRI, and biochemical testing for assessment of clinical phenotype. The study aimed to assess vitamin D levels and bone metabolism in NF1 patients, analysing potential correlations with clinical phenotype.
Studies involving physical examination								
1	Arigon [63]	2002	France (>USD 13,205)	-	32 (14)	232	Ophthalmologic complications	A cohort study of NF1 patients who received an ophthalmologic examination. The study aimed to evaluate the usefulness of ophthalmologic examination for the diagnosis and detection of complications in adult patients with neurofibromatosis type 1.
2	Khosrotehrani [64]	2003	France (>USD 13,205)	Blood pressure measurement	33 (13)	378	Multiple manifestations (cutaneous, neurofibromas, Lisch nodules, skeletal abnormalities, hypertension)	A cohort study of NF1 patients who received a full clinical examination. The study aimed to identify the main clinical features associated with mortality in NF1 patients.
Studies involving CT								
1	Nishida [65]	2016	Japan (>USD 13,205)	MDCT	47 (13)	118	GISTs	A cohort study of NF1 patients who received screening, and of NF1 patients surveyed in Japan who underwent MDCT. The study aimed to determine the risk, clinicopathologic features, and prognosis of NF1-GIST.

NF1, Neurofibromatosis Type 1. WBMRI, whole body magnetic resonance imaging. WSMRI, whole spine magnetic resonance imaging. FDG PET-CT, fluorodeoxyglucose positron emission tomography computerised tomography. MPNST, malignant peripheral nerve sheath tumour. BPNST, benign peripheral nerve sheath tumour. DXA, dual X-ray absorptiometry. CNS, central nervous system. MDCT, multi-detector computerised tomography. DWI, diffusion weighted imaging. OPG, optic pathway glioma. ADC, apparent diffusion coefficient. SUVmax, maximum standard uptake value.

**Table 2 cancers-16-01119-t002:** Summary of socioeconomic status, NF1 complication assessed, and diagnostic modalities of included studies.

Country	Studies (*n*)		NF1 Complication Assessed, *n* (%)					Modalities (*n*)	Modality, *n* (%)					
		Tumour Burden	MPNST	BMD	OPG	Spinal Abnormalities	Others		MRI	FDG PET/CT	CT	Physical Examination	Blood Pressure Measurement	DXA
All countries	51	8 (15.7)	23 (45.1)	3 (5.9)	2 (3.9)	5 (9.8)	10 (19.6)	67	34 (50.7)	16 (23.8)	5 (7.5)	7 (10.4)	2 (3.0)	(4.5)
Germany	14	1 (7.1)	8 (57.1)	1 (7.1)	1 (7.1)	1 (7.1)	2 (21.4)	18	8 (44.4)	6 (33.3)	1 (5.6)	2 (11.1)	-	1 (5.6)
United States	14	6 (42.9)	5 (35.7)	-	-	2 (14.3)	1 (7.1)	17	12 (70.6)	4 (23.5)	1 (5.9)	-	-	-
France	8	-	5 (62.5)	-	1 (12.5)	-	2 (25)	12	3(25)	3 (25)	2 (16.7)	3 (25)	1 (8.3)	-
United Kingdom	7	-	2 (28.6)	-	-	2 (28.6)	3 (42.9)	10	6 (60)	2 (20)	-	1 (10)	1 (10)	-
Japan	4	-	3 (75)	-	-	-	1 (25)	4	2 (50)	1 (25)	1 (25)	-	-	-
Finland	1	-	-	1 (100)	-	-	-	1	-	-	-	-	-	1 (100)
Italy	1	-	-	1 (100)	-	-	-	3	1 (33.3)	-	-	1 (33.3)	-	1 (33.3)
Belgium	1	1 (100)	-	-	-	-	-	1	1 (100)	-	-	-	-	
Türkiye	1	-	-	-	-	-	1 (100)	1	1 (100)	-	-	-	-	-

Abbreviations: MPNST, malignant peripheral nerve sheath tumour. MRI, magnetic resonance imaging. FDG PET/CT, fluorodeoxyglucose positron emission tomography/computerised tomography. BMD, bone mineral density. OPG, optic pathway glioma.

**Table 3 cancers-16-01119-t003:** Diagnostic accuracy of 18-FDG PET for assessing whether peripheral nerve sheath tumours are malignant or benign.

Authors	Number of Patients	Number of Lesions (MPNST)	Quantified Parameter	Threshold Value	Mean SUVmax		Sensitivity (%)	Specificity (%)
					MPNST	Benign NF		
Brenner [47]	16	N/A (16)	SUV	3.0	5.7	N/A	75	100
Chirindel [48]	41	93(24)	SUL	3.2	6.5	2.0	91	84
Salamon (2013) [50]	50	159 (19)	SUV	3.5	8.4 ±3.2	2.6 ±1.2	100	68.9
Salamon (2015) [51]	18	74 (19)	SUV	3.5	10.3 ± 4.2	4.2 ± 1.6	94.1	75
Salamon (2014) [52]	49	31 (18)	SUV	3.5	8.61	2.56	100	79.8
Van der Gucht [53]	49	40 (16)	SUV	4.5	8.8	2.9	94	88
Bredella [54]	45	50(24)	SUV	3.0	8.5 ± 0.63	1.5 ± 0.37	95	72
Nishida [55]	36	57(14)	SUV	4.1	7.43 ± 1.84	4.575 ± 1.69	92.9	88.9
Combemale [56]	113	145 (41)	T/L Ratio	1.5	N/A	N/A	97	76
Ahlawat [57]	21	55 (19)	SUV	3.2	8.0 ± 3.9	3.2 ± 1.8	100	83
Ferner [58]	18	20 (5)	SUV	2.5	5.4 ± 2.4	1.54 ± 0.7	N/A	N/A
Broski [59]	38	43 (20)	SUV	3.0	10.1 ± 1.0	4.2 ± 0.4	90	78

Abbreviations: MPNST, malignant peripheral nerve sheath tumour. SUVmax, maximum standardised uptake value. NF, neurofibroma.

**Table 4 cancers-16-01119-t004:** Bone densitometry analysis of NF1 patients.

	Number of Patients	Normal BMD, *n* (%)	Osteopenia (BMD between 1.0 and 2.5 SD), *n*(%)	Osteoporosis (BMD ≤ 2.5 SD), *n* (%)
Heerva [61]	19	6 (31.6)	5 (26.3)	8 (42.1)
Seitz [60]	14	-	6 (42.9)	8 (57.1)
Modica [62]	24	12 (50)	9 (37.5)	3 (12.5)

Abbreviations: BMD, bone mineral density.

**Table 5 cancers-16-01119-t005:** Comparison between various national guidelines.

	ACMG Guidelines [4]	French Guidelines [6]	ERN GENTURIS Guidelines [5]	United Kingdom Guidelines [7]
Tumour burden	Consider baseline MRI of known or suspected nonsuperficial plexiform neurofibromas	Whole-body MRI of NF1 patients allows assessment of the burden of internal neurofibromas. However, screening MRI for the detection of an internal neurofibroma or MPNST is not systematically recommended in patients with NF1.	Imaging by whole body MRI to monitor for plexiform neurofibromas should be performed at least at transition from childhood to adulthood to evaluate internal tumour burden as a predictor for the development of malignant peripheral nerve sheath tumour risk. (Level of evidence: Weak)	Baseline brain and spine MRI, and routine imaging of the chest and abdomen to identify asymptomatic tumours, do not influence management and should not be undertaken.
MPNSTs	Clinical suspicion (pain, rapid growth, neurologic symptoms, deep, truncal location of PN), and awareness of known risk factors (germline microdeletion of the NF1 locus, previous radiation) remain paramount for early detection of MPNST, which is facilitated by targeted MRI imaging.	The most sensitive and specific noninvasive indicator of malignant potential is [18F]2-fluoro-2-dexoy-D-glucose positron emission tomography-computed tomography using visual assessment and semiquantitative assessments with a cutoff SUV.	When clinical signs and symptoms point towards malignancy (suspicious tumours), investigation should begin with regional MRI. Prior to surgery, MRI should be carried out and 18FDG PET MRI (preferred) or 18FDG PET CT (if 18FDG PET MRI is not available) undertaken, using visual assessment and semiquantitative assessments with a cutoff standardised uptake value. (Level of evidence: Moderate)	Fluorodeoxyglucose positron emission tomography allows the visualisation and quantification of glucose metabolism in cells and is a useful diagnostic tool in differentiating benign plexiform neurofibromas from MPNST.

Abbreviations: MRI, magnetic resonance imaging. NF1, Neurofibromatosis Type 1. SUV, standardised uptake value. MPNST, malignant peripheral nerve sheath tumour. 18 FDG PET, 18 fluorodeoxyglucose positron emission tomography. CT, computerised tomography. ACMG, American College of Medical Genetics and Genomics. ERN GENTURIS, European Reference Network for all patients with one of the rare genetic tumour risk syndromes.

## Data Availability

Protocol, template data collection forms, data extracted from included studies, data used for analyses, and analytic code available from the corresponding author on reasonable request.

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
