# Peer review of "A Systematic Review of Diagnostic Modalities and Strategies for the Assessment of Complications in Adult Patients with Neurofibromatosis Type 1"

_cancers, 2024, doi:10.3390/cancers16061119_

Round 1

Reviewer 1 Report (Previous Reviewer 4)

Comments and Suggestions for Authors

The current version of the manuscript addressed my concerns. 

Author Response

We thank the reviewer for his/her comments.

Reviewer 2 Report (New Reviewer)

Comments and Suggestions for Authors

The authors present the results of a literature search on the "surveillance" of adult patients with neurofibromatosis type 1. The authors selected for published literature in English from 2000 to present day.  Initially identified over 2600 papers, only 51 included in the final paper.  The authors describe the surveillance reported in each study and demonstrate a lack of published literature on NF1 adult surveillance from low and middle socio-economic countries.  

1.  The use of the term surveillance is a bit misleading for the data and conclusions in this manuscript.  For instance, the authors report on several publications which discuss the use of PET/CT imaging for the diagnosis of MPNSTs.  PET/CT imaging is not recommended as an ongoing monitoring or imaging method in patients without suspected malignant transformation (ie baseline or surveillance).  Several of the brain tumor papers were looking at potential predictive factors in patients with known CNS lesions undergoing monitoring.  I think a better way to describe the work and conclusions here would be the published use of techniques and imaging for diagnosis of potential NF1 associated complications. The use of surveillance is confusing to the reader.  For instance, line 61 - a change from surveillance to diagnostic would be clearer. 

2.  The data under modalities (starting line 136) could be more clearly summarized in a table or chart - to include the modality, manifestation assessed for, number of identified papers, etc. This is partially done in table 2, but could be clarified. 

3.   Table 1 would be more effective if it was grouped either by NF1 manifestation assessed (ie MPNST, PNs, Bone) or by monitoring modality (MRI, PET, physical), not in alphabetical order (title says year, but not done by year). See above, would improve clarity. 

4.  The lack of inclusion of non-English published literature is a clear limitation, as this manuscript aimed to assess the care in lower socioeconomic countries.  It would be important to present the number of papers excluded by this criteria.  

5.  Table 6 is interesting in both the common recommendations or assessment of strength of evidence.  This should be commented in the discussion.  

6.  The spacing on table 2 should be improved. Study type might be better listed as NF1 manifestation assessed. 

7.  Table 3 does not significantly add to the paper given the high number of missing data points (ie nothing but percentage of tumor from Plotkin et al).  This table should be condensed or removed. 

Minor changes: 

Line 57 is quite general and needs a reference.  Adult concerns clearly include malignancy and cutaneous features.  Bone and ocular less clear.  

Line 39 states a lack of data from both high and low, but reports from high income countries are presented here.  Please clarify this line.  

Line 64/66 discuss baseline imaging and include both MRI and PETCT.  The presented data utilizes PETCT much more for diagnosis of potential malignancy.  The wording on this section is difficult to follow and most NF physicians/recommendations would not agree with the use of PETCT as a baseline. 

Line 205 in the discussion states that none reported surveillance strategies.  This is not clear given the data presented.  Full surveillance strategies? recommendations? Please clarify. 

Line 260 in the discussion about fee for service in the US is conjecture and not supported.  more access and insurance are supported by the presentation.  

Comments on the Quality of English Language

Period needed on line 125.  Malignant peripheral nerve sheath tumors do not need each letter capitalized when written out.  A few other typos noted. 

Author Response

Reply attached below

Round 2

Reviewer 2 Report (New Reviewer)

Comments and Suggestions for Authors

Overall, the authors have addressed the prior recommendations. The adjustments of wording to imaging and diagnostic modalities improves the reading. 

Comments on the Quality of English Language

No significant concerns.  Minor adjusts might improve read-ability. 

This manuscript is a resubmission of an earlier submission. The following is a list of the peer review reports and author responses from that submission.

Round 1

Reviewer 1 Report

Comments and Suggestions for Authors

This manuscript aims to detect differences in surveillance of NF1 patients in countries of different socioeconomic rank, however the methods instead highlight research and publication differences across different countries without detailing differences in practice or outcomes for NF1 patients. There is speculation that MRI and PET are less readily available in low economic income countries, which is most likely true, but there is no review of national or international guidelines for surveillance of each country, or for mortality rate differences for NF1 patients compared with the general population across the different economic categories. There is a focus on the clinic in Singapore, particularly vis-a-vis genetic testing, but genetic testing is frequently unnecessary for NF1 patients and their families, as the penetrance is 100% and stigmata are usually readily identifiable. The conclusions drawn are not supported, that published research studies can be taken as a proxy for clinical practice in different countries. They cannot.  Overall, the focus of this manuscript should be placed back on the question proposed, which is an interesting question. What ARE the surveillance modalities used in different countries for NF1 patients, and do outcome measures such as survival and incidence of malignancy conform to clinical practice? This paper, however, does not directly address that question.

Author Response

Changes to the original text have been highlighted with track-change in the file "Manuscript 121123 with tracked changes"

Reviewer 2 Report

Comments and Suggestions for Authors

The surveillance plays a critical role in all inherited tumor predisposition syndromes.

Therefore, the aim of the paper, to review the international surveillance strategies and analyze a possible association between the socioeconomic status of the country and the surveillance strategies used, is an actual and crucial issue. Although a comprehensive analysis has been done, some major points should be addressed as follows: Fifty-one papers have been selected using a correct search strategy Fifty studies were conducted in high-income countries and one was conducted in an upper-middle income country. The material collected does not allow to make further analysis. The only possible conclusion as the Authors wrote is “ that there is robust data on surveillance techniques for adult NF1 patients in high-income countries, but not for lower- income countries” There are differences between the various surveillance protocol adopted in high income countries but they were not examined and discussed in the paper

Discussion part is very light a long part has been dedicated to describe the NF1 patient management in Singapore a very interesting subject to consider, but it does not fit within the aim of the paper.  

Minor point

57 “As  patients mature to adolescence and adulthood, several complications arise, such as an increased risk of malignancy, cutaneous, ocular, orthopedic, vascular, and neuropsychiatric symptoms”. The sentence should be corrected, Neuropsychiatric symptoms are often present since the childhood.

Comments on the Quality of English Language

editing of english language is required  because the are unclear phrases and grammar mistakes

Author Response

Please see the attachment. Changes to the original text have been highlighted with track-change in the file "Manuscript 121123 with tracked changes"

Reviewer 3 Report

Comments and Suggestions for Authors

The manuscript attempts to systematically review the literature on published surveillance strategies in adults with NF1.  Although this is an interesting topic, a number of significant concerns are apparent in the manuscript and should be addressed.  

1) In Aims of the paper, authors describe “surveillance strategies for NF1 patients.” However, the search terms used are largely oriented to imaging modalities for use in tumor surveillance.  NF1 is a complex disease with multiple manifestations, but many surveillance modalities (quality of life surveys, mental health screening, adaptations to the work environment, socioeconomic status) would not be included in the analysis because they were not considered in search terms. As a result, this review is not systematic or inclusive.  Instead, the authors do a good job collecting studies of imaging surveillance.  They could limit their analysis to these studies if some additional studies (such as those limited to physical exam or ophthal evaluation) were excluded.  I worry that if these non-imaging studies are included, then the manuscript as a whole will be very incomplete.

2) Differences between studies and modalities are not investigated.  For instance, Ferner et al performed their PET scans 4 hours after injection which increases SUV uptake compared with more traditional nuclear imaging methods.  Although studies on the same subject are presented as equivalent, they commonly have differences in image acquisition or processing that are important to describe.

3) This study investigates published trials of (mostly imaging) surveillance for malignancy in adults with NF1. This is not the same as published recommendations for surveillance, and the authors dedicate a paragraph in the discussion to reviewing recently published imaging recommendations from large groups.  This is important for readers to take away from this paper, and I would suggest emphasizing these group recommendations by displaying them in a table.

4) This paper investigates the country of origin for trials investigating the clinical utility of imaging screening.  Although no studies were found from low or low middle income countries, that does not mean that these countries are not employing screening strategies.  To investigate this, authors would have to examine real world evidence from those countries and not costly published articles investigating experimental procedures.  This calls into question many of the conclusions of this paper which should be revised. 

The manuscript has a significant amount of space dedicated to the landscape in Singapore and the development of an adult NF clinic.  Although this is clearly important to the authors, it is of limited use to readers.  Space in this manuscript could be better used describing surveillance techniques in NF1.

5) The authors seem to miss an important opportunity to discuss relative risk of different imaging modalities.  MRI is generally preferred over CT due to the use of radiation that increases the risk of malignancy.  PET/CT is often used for MPNST surveillance but also requires radiation.  This is an important aspect in determining potential surveillance strategies in lower income countries.

Other more minor points to be addressed:

1) How were studies in other languages handled since this may bias toward studies in certain languages?

2) For studies by Ina Ly, her last name is “Ly” and she commonly goes by the first name “Ina.” So, if referring to her last name, these studies would be better called “Ly”.

3) The study is better termed as a review than as a systemic review since there are numerous limitations in systemic review methodology and a lack of combined systemic analysis. 

Author Response

(The authors gave the same response as above.)

Reviewer 4 Report

Comments and Suggestions for Authors

The authors performed a comprehensive literature review of neurofibromatosis type 1 (NF1) directed surveillance strategies.  Based on the research publications, they arrive to the conclusion that there is limited use of imaging modalities in low-income countries.  

Medical imaging, especially advanced techniques such as functional MRI and PET-CT are expensive and no one would argue that access to these procedures is equally available throughout the world, however, I don’t think that research publications are the right data source to prove that inequality.   

Many of the publications included in the review present data from research studies that compare imaging modalities with the aim to identify the best diagnostic techniques for specific aspects of NF1 and therefore not indicative of the everyday practice.  Some aspects of established NF1 specific surveillance strategies, for example mammography do not come up in the literature search focused on cohort studies despite the recommendation to start annual mammography exams in NF1 patients from age 30 being included in the consensus guidelines.  Certain elements that are part of the medical evaluation, such as physical exam, blood pressure measurement, ophthalmic or neurological exams may not be reported in research papers that focus on imaging, but that does not mean those exams are not routinely performed.  Similarly, additional imaging that was not relevant to the research topic may not be mentioned. 

In my view, the fact that research papers were not published based on populations of low-income countries neither proves nor disproves the utilization of any medical procedures in those countries.

The literature review by itself is fascinating and there are many logical conclusions that can be made.  I would encourage the authors to take a fresh look at the data and find a message that can be directly derived from the facts.

Author Response

(The authors gave the same response as above.)

Round 2

Reviewer 1 Report

Comments and Suggestions for Authors

The edits submitted have greatly improved the merit of this paper. I still observe a disconnect between the stated goals of the paper, and the methods utilized to address these goals, however the Discussion section does much to clarify the overall importance of the investigation. I have several remaining suggestions that would satisfy these concerns:

1) It should be indicated in the introduction/methods that utilizing published research to estimate common practice in other nations assumes that there is a direct correlation between publications utilizing surveillance strategies and the clinical use of surveillance strategies. This is not a given.

2) Also, it should be clearly noted that exclusion of non-English language papers was (likely-- I am guessing) necessary due to difficulty in interpreting those publications by scientists who are not fluent in those languages, however their exlusion possibly obscures rare but informative data from underrepresented countries. Also, it may help to list the number of non-English publications excluded from analysis. 

3) The following is a suggestion but is not required: There are national and international foundations who could be called upon to derive treatment recommendations based on GNI -- a stratified and streamlined list of recommendations. The Children's Tumor Foundation may be the largest such group. They could pair with smaller regional Foundations knowledgeable about the common practices of each nation to derive recommendations that make sense for each unique country and culture. One of the reasons I mention this is that auditory testing was recommended in one of your citations and this does not make much sense to me for NF1 patients who suffer from pediatric optic pathway gliomas (OPGs) but not vestibular nerve tumors. They could also probably add more value by providing more specific and helpful recommendations than just "annual physical and skin exam". 

4) The term used in the U.S. is Neurofibromatosis Type 1, or NF1. I have not seen prior publications refer to it as Neurofibromatosis 1. If you have information to contradict this observation, by all means feel free to keep the terminology, but I believe at this time that it would be more accurate to use the terminology as mentioned above. 

Author Response

The response is attached.

Reviewer 3 Report

Comments and Suggestions for Authors

The authors have submitted a response to reviewer comments.  Changes include a new table and a more focused discussion, which are appreciated.  However, other changes that were mentioned in the authors' cover letter do not appear in the revised manuscript.  This includes a revision of the title and the exclusion of studies that are not focused on imaging.  Unfortunately, other changes (such as the clarification that the manuscript was restricted to English only journal articles) also limit its impact.  In the end, I worry that this manuscript points out that journals written in English commonly publish articles on imaging surveillance in NF1 and that non-English journals often do not publish these articles.  Although there may be some merit in reviewing the surveillance strategies of various high-SES countries in NF1 and potentially editorializing that adapting these strategies for lower-SES countries would be important, the authors seem less excited by this strategy. 

Author Response

The response is attached.

Reviewer 4 Report

Comments and Suggestions for Authors

Based on the literature review of neurofibromatosis type 1 (NF1) directed surveillance strategies, the authors in the previous version of the manuscript came to the false conclusion that the lack of research publications originating from low-income countries reflects the limited use of imaging modalities in those countries.  In the updated version, the conclusion is that there is a need to develop clinical practice guidelines that are applicable to low-income countries.  There is a leap from imaging focused surveillance to clinical practice guidelines, but the statement is not factually untrue.

One remaining issue: the manuscript seems to conflate surveillance technique with surveillance strategy.  The majority of research publications grouped together as surveillance are testing whether one modality or technique is superior to another in detecting a certain complication, such as the development of malignancy. Comparing the diagnostic accuracy of PET-CT versus MRI or imaging procedures with or without contract material is unlikely to generate different results based on the economic status of participants.  Whereas employing a surveillance strategy, such as regular screening exams of one sort or another would be population dependent.  Although there is an expanded discussion of this aspect in the new version, only a fraction of the cited literature actually deals with that. 

I recommend clearly separating the technical efficacy studies from the strategy focused studies.  It could be emphasized that results from the first group inform the strategy focused studies, and it is true that many of these utilize imaging techniques that have limited availability in the developing world. 

Imaging is resource intensive, but there are research studies looking at affordable ways of improving the outcome of patients with NF1, that do not include imaging, however, these were missed by the literature search that focused heavily on the use of imaging modalities.

Author Response

The response is attached.
